# Treatment Advances for Acne Vulgaris: The Scientific Role of Cannabinoids

Inês Ferreira [1,2] , Carla M. Lopes [1,2,3,*] and Maria Helena Amaral [1,2,*]

1 UCIBIO—Applied Molecular Biosciences Unit, MedTech—Laboratory of Pharmaceutical Technology, Faculty of Pharmacy, University of Porto, Rua de Jorge Viterbo Ferreira 228, 4050-313 Porto, Portugal; inesdani2000@hotmail.com
2 Associate Laboratory i4HB, Institute for Health and Bioeconomy, Faculty of Pharmacy, University of Porto, Rua de Jorge Viterbo Ferreira 228, 4050-313 Porto, Portugal
3 FP-I3ID—Instituto de Investigação, Inovação e Desenvolvimento, FP-BHS—Biomedical and Health Sciences Research Unit, Faculdade de Ciências da Saúde, Universidade Fernando Pessoa, Rua Carlos da Maia 296, 4200-150 Porto, Portugal
* Correspondence: cmlopes@ufp.edu.pt (C.M.L.); hamaral@ff.up.pt (M.H.A.)

**Abstract:** Acne vulgaris is a prevalent dermatological disorder that impacts the quality of life for millions of people around the world. The multifactorial nature of this disorder requires innovative and effective treatment strategies. Over time, there has been a growing interest regarding the use of natural topical therapies, with cannabinoids emerging as a promising group of compounds for investigation. In the context of acne treatment, cannabinoids are of particular interest due to their anti-acne properties, namely, lipostatic, anti-inflammatory, antiproliferative, and antimicrobial activities. Among these bioactive compounds, cannabidiol stands out as a notable derivative, exhibiting a promising spectrum of therapeutic actions. Pre-clinical and clinical studies have proven its ability to modulate sebum production, reduce inflammation, and inhibit bacterial proliferation—all of which are critical components in the pathogenesis of this dermatosis. This review provides a comprehensive overview of cannabinoids' potential as a novel and holistic approach to acne vulgaris treatment and summarizes recent developments in this area.

**Keywords:** acne vulgaris; cannabinoids; cannabidiol; sustainability; lipostatic; anti-inflammatory; antiproliferative; antimicrobial

## 1. Introduction

Nowadays, there has been a significant rise in concerns related to the environment and sustainability from the perspective of the circular economy. Therefore, several industries are shifting toward plant-based ingredients to explore their application in fields that require new alternatives [1]. A notable example of this trend is the expanding research in cannabis-derived products, which have gained attention in recent decades because of cannabis legalization in an increasing number of countries and further research on its potential [1]. As a result, scientists and researchers are actively investigating the diverse range of benefits associated with cannabis-derived compounds. This includes the study of cannabinoids, terpenes, and other bioactive components present in plants, which have shown promising therapeutic properties in multiple fields, such as medicine, wellness, skincare, and nutrition.

The shift toward plant-based alternatives reflects not only a response to environmental concerns but also an increasing recognition of the potential benefits that cannabis-derived compounds may offer across diverse fields [2–6]. A specific focus of this exploration is the potential use of cannabinoids for treating dermatologic disorders such as acne vulgaris. Traditionally managed with various pharmacological treatments, acne is now being studied in the context of cannabinoids, particularly cannabidiol (CBD), which shows promise in addressing inflammation and regulating sebum production [1]. This emerging area

holds the potential to reshape skincare practices and provide innovative solutions for acne management.

Furthermore, considering the significant impact of acne on individuals' lives, there has been an increase in the number of recent reviews delving into the topic of acne vulgaris [7–11]. This review aims to consolidate the latest advancements in this field and to highlight the potential of cannabinoids. It provides a comprehensive overview covering the genesis of acne, clinical evaluation, available treatments, and the associated side effects of conventional therapeutic approaches. Taking these factors into account, this review explores how cannabinoids can prove to be a viable alternative through an understanding of their origin and an analysis of the pre-clinical and clinical studies that reveal their therapeutic properties.

This review has significance through not only challenging established approaches but also through proposing a promising trajectory for future dermatological advancements, thereby contributing to the evolving landscape of acne treatment.

## 2. *Cannabis sativa*

*Cannabis sativa* (*C. sativa*) is a herbaceous plant in the *Cannabis* genus and a species of the *Cannabaceae* family that can provide an abundant supply of bioactive molecules for the manufacture of nutraceutical, pharmaceutical, and cosmetic products [2]. Cannabis' origin and/or main domestication have historically been attributed to Central and Southeast Asia, which are also believed to have played a significant role in its evolution [12–14].

Cannabis is a plant that originated in Asia's earliest agricultural communities and has been used throughout history for many different purposes, including oil, food, fibers, textiles, medicine, and religious practices [12]. Due to its resourcefulness and trade routes, cannabis spread across different regions and civilizations, becoming an integral part of several cultures and traditions [12].

Therefore, in contrast to the production of other naturally derived drugs, which tends to be concentrated in a small number of countries, cannabis is cultivated in almost every country worldwide. It can be grown successfully in a diverse range of environments, including both indoor and outdoor settings [15]. *C. sativa* production has grown significantly around the world, with a particular emphasis on the rise of indoor cultivation. Currently, the most significant development revolves around the expanding practice of indoor cultivation, which is primarily concentrated in Europe, Australia, and North America [2].

Hemp, a variety of cannabis, is extensively cultivated across Europe. In recent years, the area dedicated to hemp cultivation in the European Union (EU) has grown significantly, rising from 20,540 hectares (ha) in 2015 to 33,020 ha in 2022 (a 60% increase). During the same period, hemp production surged from 97,130 tons to 179,020 tons (an 84.3% increase). France leads as the largest producer, accounting for over 60% of EU production, followed by Germany (17%) and the Netherlands (5%) [16].

The chemical composition of cannabis varies significantly because it is influenced by multiple factors, including growing conditions (such as moisture, nutrition, and light levels), harvest timing, storage conditions, plant type, age, and variety. These factors, among others, contribute to the diverse concentrations of compounds found in cannabis [15]. The principal secondary metabolites produced by this plant include terpenes, polyphenols, and cannabinoids [2]. Terpenes are represented by more than 100 molecules that can be found in the roots, flowers, and leaves, as well as in secretory glandular hairs, which are their main source of production. This class of compounds includes monoterpenes, sesquiterpenoids, triterpenes, diterpenes, and terpenoid derivatives [15]. Additionally, the plant contains more than 20 polyphenols, the majority of which are flavonoids derived from flavone and flavanol. Among the examples of flavonoids identified are kaempferol, quercetin, apigenin, luteolin, and three prenylated/geranylated flavones, cannflavin A, B, and C [15]. The other typical chemical components isolated from cannabis include fatty acids, amino acids, sugars, and vitamins [17].

Cannabinoids are among the most abundant metabolic products of cannabis, regardless of having fewer than 20 molecules [2].

*2.1. Cannabinoids*

Cannabinoids constitute a diverse group of pharmacologically active compounds that share structural and biological similarities with the primary psychoactive compound, Δ9-tetrahydrocannabinol (Δ9-THC) [18,19]. Cannabinoids can be categorized into three major classes based on their origin: phytocannabinoids, synthetic cannabinoids, and endocannabinoids [18].

2.1.1. Phytocannabinoids

Phytocannabinoids constitute a category of terpenophenolic compounds mainly synthesized by plants within the *Cannabis* genus [20–24]. Nonetheless, these substances are not exclusive to Cannabis; they can also be extracted from other plant species, such as those in the genera *Rhododendron* and *Radula* [25]. Among these sources, *C. sativa* emerges as the most extensively studied and prevalent productive source of phytocannabinoids [25]. This plant is notable for its high concentrations and diversity of phytocannabinoids, making it a primary focus in the study of these compounds [25]. Alongside other specialized metabolites, phytocannabinoids are responsible for the typical biological activity of *C. sativa* [26]. They are primarily found in the resin secreted by female plant trichomes. This class includes over 100 compounds. The most well-known members of this class are Δ9-THC and CBD [18]. While Δ9-THC is psychoactive, CBD has no-psychoactive properties [17].

2.1.2. Synthetic Cannabinoids

Synthetic cannabinoids are developed in laboratories and have chemical and structural similarities to endocannabinoids and phytocannabinoids [19]. This group of molecules is being diversified and advanced to investigate the therapeutic potential of cannabinoids [17]. Examples of synthetic cannabinoids include WIN-55,212-2, JWH-133, (R)-methanandamide (MET), and CP 55,940 [19]. There are currently two highly publicized synthetic cannabinoids already on the market, namely, dronabinol and nabilone [20].

2.1.3. Endocannabinoids

Endogenous cannabinoids (i.e., endocannabinoids) are naturally produced by humans and other animals. These compounds are commonly recognized as neuromodulator agents. They possess distinctive features that set them apart from typical neurotransmitters: they are synthesized on demand at their site of action through receptor-stimulated cleavage of lipid membrane precursors and are not stored in synaptic vesicles [27–36]. They are arachidonic acid (AA) derivatives synthesized from phospholipids on the inner leaflet of the membrane, and the most well-known members of this class include 2-arachidonoylglycerol (2-AG) and anandamide (AEA) [19,37,38].

Table 1 lists some examples of the three classes of cannabinoids mentioned previously (i.e., phytocannabinoids, synthetic cannabinoids, and endocannabinoids).

**Table 1.** Summary of different classes of cannabinoids, adapted from Eagleston et al. [19].

| Cannabinoid Type | Members of Class |
|---|---|
| Phytocannabinoids | Δ (9)-tetrahydrocannabinol (THC)<br>Cannabidiol (CBD)<br>Cannabidiolicacid (CBDA)<br>Cannabigerol (CBG)<br>Cannabichromene (CBC)<br>Cannabinol (CBN)<br>Cannabidivarin (CBDV)<br>Cannabigerovarin (CBGV)<br>Δ (9)-tetrahydrocannabivarin (THCV)<br>Tetrahydrocannibinolic acid (THCA) |

**Table 1.** *Cont.*

| Cannabinoid Type | Members of Class |
|---|---|
| Synthetic cannabinoids | WIN-55,212-2<br>JWH-133<br>(R)-methanandamide (MET)<br>CP 55,940<br>Dronabinol<br>Nabilone |
| Endocannabinoids | 2-arachidonoylglycerol (2-AG)<br>Anandamide (AEA)<br>N-arachadonoyl dopamine<br>Homo linoleoyl ethanolamide (HEA)<br>Docosa tetraenyl ethanolamide (DEA)<br>Virodhamine<br>Noladin ether |

### 2.2. The Endocannabinoid System

Endocannabinoids carry out their functions by interacting with the endocannabinoid system (ECS), a complex intercellular signaling network responsible for maintaining homeostasis in the human body [37]. This system is composed by the following elements: endocannabinoids, which act as signaling molecules; specific receptors; enzymes that synthesize and breakdown endocannabinoids; and transporters of the endocannabinoids [19,37,39–41].

The G-protein-coupled receptors, i.e., cannabinoid type 1 (CB1R) and cannabinoid type 2 (CB2R), are the two main receptors through which endocannabinoids exert their effects [1]. Both receptors contain a glycosylated amino-terminal (extracellular) and carboxy-terminal (intracellular) domain that is connected by seven transmembrane domains, three extracellular loops and three intracellular loops [17,42,43].

Cannabinoid type 1 receptor (CB1R) and cannabinoid type 2 receptor (CB2R) are distributed differently in the body. The CB1R is highly expressed in the central nervous system and is principally linked to the psychoactive effects of cannabinoids. It is predominantly pre-synaptic in the brain and is responsible for the regulation of memory, mood, sleep, appetite, and pain sensation via the release of various neurotransmitters. The CB1R is also present in lower concentrations in peripheral tissues, including cardiac, testicular, muscular, hepatic, pancreatic, and adipose tissues, among others [19,44]. The CB2R is mostly found in the peripheral nervous system. It is primarily expressed in the spleen and in cells of hematopoietic lineage and is presumed to be responsible for the immunomodulatory and anti-inflammatory effects of the cannabinoids [19,45].

When cannabinoids bind to the CB1R/CB2R, they induce a conformational change in the receptors, transitioning them from an inactive to an active state. Consequently, the $G\alpha$ subunit of the G protein is separated from the CBR as well as the $G\beta\gamma$ dimer [1]. When activated, cannabinoid receptors interfere with various signaling pathways to exert their effects on different tissues and organs. The activation can lead to diverse effects on cell physiology depending on the type of cell involved. This activation can inhibit adenylyl cyclase and specific voltage-dependent calcium channels while also activating various mitogen-activated protein kinases (MAPK) and inwardly rectifying potassium channels. As a result, activation of the CB1R or CB2R can trigger a variety of cellular responses, such as changes in gene transcription, cell motility, and synaptic function, among other things [46]. For example, in neurons, CB1's pre-synaptic stimulation inhibits neurotransmitter release. In the liver, where CB1 expression is normally low, its stimulation increases acetyl-Coenzyme A carboxylase and fatty acids, resulting in increased lipogenesis [46,47]. On the other hand, activation of the CB2R, which is mostly expressed in immune system cells, seems to mediate immunosuppressive effects [46,48].

2.2.1. The Endocannabinoid System of the Skin

The skin is the human body's largest organ and the first line of defense against pathogens and potential damage caused by chemicals, biological agents, and UV radiation. It acts as a protective barrier that separates the external environment from the internal body [1].

Recent studies have reported the presence of a unique ECS within the skin. This discovery is supported by the identification of endogenous ligands for the CB1R and CB2R in the skin [49]. The epidermal ECS actively contributes to skin homeostasis and skin barrier integrity, with endocannabinoids intricately involved in regulating various neuro-immunoendocrine functions of the skin [1,50].

The two main cannabinoid receptors, CB1R and CB2R, have been identified in dermal cells, melanocytes, epidermal keratinocytes, mast cells, sweat glands, hair follicles, and cutaneous nerve fibers [1,51]. In addition, other skin receptors targeted by cannabinoids, such as transient receptor potential (TRP) ion channels and peroxisome proliferator-activated receptors (PPARs), have been detected in multiple skin cells. For example, transient receptor potential vanilloid-4 (TRPV4) was found in the secretory cells of eccrine sweat glands, TRPV1 was found in keratinocytes, and TRPV3 was found in hair follicles [52–54]. Among PPARs, PPARα was shown to be highly expressed in the epithelial compartment of basal keratinocytes and hair papilla cells, and PPARδ and PPARγ were shown to be highly expressed in the epidermis, hair follicle, and sebaceous glands [52,55–57].

Moreover, the existence of enzymes engaged in the synthesis and degradation of endocannabinoids within the skin suggests that the cutaneous system is also actively engaged in cannabinoid metabolism. Notably, sebocytes, melanocytes, fibroblasts, and immunocytes have been found to contain specific enzymes, like fatty acid amide hydrolase (FAAH) and monoacylglycerol lipase (MAGL) [1].

Regarding the presence of endocannabinoids in the skin, two extensively studied compounds are 2-arachidonoyl glycerol (2-AG) and anandamide (N-arachidonoyl ethanolamide, AEA). These endocannabinoids have garnered significant research attention due to their abundance and importance in skin-related studies [37].

2.2.2. Therapeutic Potential of Cannabinoids in Dermatology

Following the discovery of a skin ECS, extensive research has been conducted to understand its role in the functioning of this organ and how disruptions in its normal operation can lead to the development of pathological skin disorders [58]. The ECS plays a crucial role in regulating important aspects of cutaneous function, such as cell growth, differentiation, survival, immune and inflammatory responses, as well as sensory phenomena [59].

The ECS's multiple anti-inflammatory and immunomodulatory properties have been documented through various studies, which have utilized cannabinoid receptors, selective agonists, antagonists, and other regulatory agents to effectively regulate the levels and actions of ECBs during inflammatory processes, thereby providing robust evidence for the ECS's therapeutic potential [59–64].

Cannabinoids exhibit significant potential in the field of dermatology due to their anti-inflammatory, antipruritic, and antinociceptive properties [56]. Extensive research has consistently reported the effectiveness of these compounds in treating several skin disorders, including allergic contact dermatitis, eczema, pruritus, psoriasis, skin cancer, acne vulgaris, and others [19,37,38,65,66]. This review focuses specifically on the potential use of cannabinoids as a therapeutic alternative for controlling acne.

## 3. Acne Vulgaris

Acne vulgaris is a highly prevalent skin condition, with a global incidence of 9.38% across all age groups [8]. It is ranked as the eighth most prevalent skin disorder in the world [8]. The incidence of this pathology differs in multiple countries and age groups, with estimates ranging from 35% to nearly 100% of adolescents experiencing acne at some

stage. Acne is prevalent among nearly all individuals aged 15 to 17, with a significant proportion (around 15–20%) suffering from moderate to severe forms of the condition [8].

The initial development of acne typically begins with the onset of puberty, when sebum production increases. Consequently, the prevalence of acne rises with age, peaking in adolescence and declining in prepubescent children. After achieving late adolescence or early adulthood, the incidence of acne follows a decreasing trend with age [67,68]. However, recent research has indicated an increase in adult acne occurrence, particularly among women [8,69].

Acne prevalence is determined by skin type, and genetic factors. In order to reduce its prevalence, it is crucial to raise people's awareness of skin care, variables that can cause acne, and available treatments [69].

*3.1. Pathogenesis*

Acne vulgaris, or simply known as acne, is a chronic and multifaceted skin condition that primarily affects the face, upper back, and upper chest [70]. It is a complex condition whose genesis and evolution can be triggered by an interplay of multiple factors, including stress, nutritional, and hormonal alterations [71,72]. Acne is a pathologically inflammatory disease that mainly affects the pilosebaceous units. These units are complex structures of hair follicles, hair shafts, and sebaceous glands [71].

The development of this dermatosis is mainly triggered by a dysregulation in the activity of the sebaceous glands, which can result in hypersecretion of sebum, irregular shedding of skin cells (hyperkeratinization), colonization by *Cutibacterium acnes* (*C. acnes*) within the pilosebaceous units, and subsequent inflammation [1,71].

Depending on the severity of the condition, this skin disorder is characterized by various symptoms, including scaly red skin (seborrhea), whiteheads (closed comedones) and blackheads (open comedones), papules, pustules, nodules, pimples, and scarring [73]. The presence of comedones indicates a non-inflammatory stage of acne, whereas lesions like papules, pustules, and nodules point to an inflammatory stage of the disease [74].

Comedones are structures that develop when the pilosebaceous units become obstructed by sebum accumulation. Blackheads are follicular plugs that form when the hair follicle remains open. When trapped sebum, bacteria, and dead skin cells oxidize, upon exposure to air, resulting in a dark or black appearance. Open comedones have a wider opening, allowing the oxidized material to be visible on the skin's surface. On the other hand, whiteheads are follicular plugs that form when the hair follicle opening becomes closed or blocked. The trapped sebum, bacteria, and dead skin cells remain beneath the skin's surface, leading to a white or flesh-colored bump [73]. The formation of these structures (comedogenesis) is linked to the influence exerted by androgen hormones since sebaceous glands are very sensitive to them. When androgen levels, like testosterone, increase significantly, the sebaceous glands become hyperstimulated, resulting in excessive sebum production [70,71]. The distension of these lesions can cause follicular rupture, leading to the development of inflammatory lesions such as papules, pustules, nodules, or cysts [75].

Follicular hyperkeratinization is also an impactful phenomenon that happens when the cutaneous cells lining the sebaceous follicle, primarily keratinocytes, fail to properly shed and detach from one another. This causes excessive cohesion or stickiness between the keratinocytes, leading to the accumulation of keratin within the hair follicle. Consequently, the excess keratin contributes to the formation of plugs or comedones that block the follicles, disrupting the continuous cycle of skin cell renewal and replacement [75,76].

Colonization of the sebaceous follicle by *C. acnes* is also a factor that can contribute to the development of comedones and the subsequent appearance of acne. *C. acnes* is a Gram-positive anaerobic bacteria, formerly designated as *Propionibacterium acnes*, that naturally inhabits the skin's microbiota and contributes to skin homeostasis [73]. Nonetheless, certain factors can influence its activity and contribute to acne development. Overproduction of sebum creates an oxygen-deprived and nutrient-rich environment that stimulates the

overgrowth and proliferation of *C. acnes*. The abundance of lipids and other substances in sebum serves as a source of nutrients for bacteria. This causes an overabundance of bacterial growth byproducts, which can stimulate the immune system, triggering the release of inflammatory mediators [71,76].

Inflammatory mediators include inflammatory cells, such as lymphocytes and neutrophils, as well as the subsequent inflammatory chemicals released by these cells, like tumor necrosis factor-alpha (TNF-α) and reactive oxygen species (ROS). These inflammatory mediators contribute to acne inflammation and progression [71]. Inflammation exacerbates the redness, swelling, and discomfort associated with acne lesions [71].

The severity of this disease is often categorized into different levels based on the number, morphology, and distribution of lesions, as well as the extent of inflammation and scarring [77]. However, there is no universally established grading system to rate acne [78].

### 3.2. Current Treatments for Acne

Acne treatment options vary depending on its severity and patients age range, treatment preferences, compliance, and reaction to prior treatments. The objective of the treatment is to minimize the severity and recurrence of skin lesions while improving their overall appearance [79,80].

The acne treatments can differ by targeting different factors involved in the pathogenesis of this skin condition, from reducing androgens and sebum production to preventing blockage of hair follicles, decreasing inflammation, and inhibiting the proliferation of *C. acnes* [81]. According to the severity of the condition, there are different treatment options (Table 2) and two sequential approaches to the disease. Moreover, acne treatments are classified as topical and systemic [81].

**Table 2.** Summary of treatment options for each severity level of acne vulgaris (mild, moderate, severe), adapted from Kraft and Freiman [81].

| | Treatment Options | |
|---|---|---|
| **Severity—Clinical Findings** | **First Line** | **Second Line** |
| Mild Comedonal | Topical retinoid | Alternative topical retinoid Salicylic acid washes |
| Papular/pustular | Topical retinoid Topical antimicrobial: ○ benzoyl peroxide ○ clindamycin ○ erythromycin Combination products | Alternative topical retinoid plus, alternative topical antimicrobial Salicylic acid washes |
| Moderate Papular/pustular | Oral antibiotics: ○ tetracyclines ○ erythromycin ○ trimethoprim ○ sulfamethoxazole Topical retinoid ± benzoyl peroxide | Alternative oral antibiotic Alternative topical retinoid Benzoyl peroxide |
| Nodular | Oral antibiotic Topical retinoid ± benzoyl peroxide | Oral isotretinoin Alternative oral antibiotic Alternative topical retinoid Benzoyl peroxide |
| Severe | Oral isotretinoin | High-dose oral antibiotic Topical retinoid (also maintenance therapy) Benzoyl peroxide |

### 3.2.1. Topical Therapy

Topical therapy is considered the standard treatment option for mild to moderate acne. The agents implied in this therapy are directly applied to the skin in the form of creams,

gels, emulsions, or ointments specifically targeting the affected areas. Acne topical therapy mainly focuses on the use of bactericidal agents and retinoids [81].

The bactericidal agents and retinoids can be used individually or in combination (known as combination therapy) to enhance the treatment effectiveness, depending on the clinical case. Nonetheless, it is important to apply them at different times unless there is evidence that they are compatible [81].

Retinoids

Topical retinoids, which are vitamin A derivatives, possess an important role in acne treatment due to their ability to minimize visible lesions while also preventing the formation of microcomedones and new lesions. Such treatment reduces inflammatory lesions and comedones by 40% to 70% [81]. Retinoids target the process of desquamation by normalizing it through a decrease in keratinocyte proliferation and promoting the process of differentiation. They also suppress various crucial inflammatory pathways that become activated in acne, including the activator protein 1 (AP-1) pathway, migration of leukocytes, and toll-like receptors [82].

There is currently a wide range of topical retinoid formulations and concentrations, which allows clinicians to create personalized treatment plans based on the presentation, preferences, and tolerance of the patient [82]. Tretinoin, adapalene, and tazarotene are the most widely used topical retinoids [81].

Antimicrobials

Acne's inflammatory lesions can be effectively treated with topical antimicrobials. Among them are benzoyl peroxide and antibiotics [81]. Benzoyl peroxide is an effective antibacterial agent for decreasing the concentration of *C. acnes* by releasing free oxygen radicals while also preventing microbial resistance [77]. Moreover, it exhibits anti-inflammatory properties and demonstrates a rapid response during the treatment; it can produce results as early as five days [81]. Benzoyl peroxide is available in different dosage forms, such as creams, topical washes, foams, or gels, and can be used as a leave-on or wash-off agent [77].

Salicylic acid is also a popular bactericidal agent. It is an effective first-line treatment for very mild acne with a reasonable safety profile. However, it is considered to be less effective than a topical retinoid [77].

Regarding the classes of antibiotics used, they may include clindamycin and erythromycin. These antibiotics are well tolerated by most individuals, and they can reduce the concentration of *C. acnes*. In several randomized, controlled trials, they have been shown to decrease inflammatory lesions by 46% to 70% [83–86]. To optimize their efficacy and minimize the development of bacterial strains resistant to treatment, these antibiotics are commonly combined with benzoyl peroxide (either as a wash-off or leave-on agent) [8]. Among topical antibiotics, clindamycin 1% gel or solution is presently recognized as one of the best options for the treatment of acne [77].

3.2.2. Systemic Therapy

Systemic treatments are typically prescribed for more severe cases of acne or when topical treatments have proven ineffective. These treatments are taken orally or administered through injections, allowing the medication to circulate throughout the body. Oral antibiotics, hormonal therapies, and isotretinoin are the primary systemic treatment options widely employed in the management of acne [81].

Oral Antibiotics

Systemic antibiotics are the second approach to treatment when topical therapy is insufficient or not well tolerated. They are particularly recommended for moderate-to-severe acne situations, mainly when the affected areas include the face, chest, and shoulders.

The response to these agents, which may include compounds such as tetracyclines, doxycycline, and minocycline, is noticeable after a minimum of six weeks of therapy. In

terms of efficiency, doxycycline and minocycline are generally recognized as more effective than tetracycline [81,87].

Hormonal Therapies

Hormonal therapies can be beneficial for individuals who have hormonal imbalances that contribute to acne. Hormonal agents, such as combined contraceptive pills, are commonly recognized as a second option of treatment for adult and adolescent women. While it may take longer for patients to observe clinical improvement, these agents have demonstrated comparable effectiveness to oral antibiotics in managing inflammatory lesions among adult women with acne [74]. Moreover, clinical evidence suggests that hormonal therapies yield particularly favorable responses in treating deep-seated nodules on the lower face and neck [81].

The anti-androgenic effects of combined oral contraceptives are the basis for their mechanism of action in the treatment of acne. These therapeutic agents work by decreasing androgen production at the ovarian level and increasing the amount of sex hormone-binding globulin, which attaches to free circulating testosterone. As a result, less testosterone is available to interact with and activate the androgen receptor. In addition, combined estrogen–progestin oral contraceptives (COCs) minimize 5-alpha-reductase activity and block the androgen receptor, which contributes to their therapeutic effects in the treatment of acne [77].

The Food and Drug Administration (FDA) has currently approved four COCs for acne treatment. These include Ortho Tri-Cyclen (ethinyl estradiol–norgestimate), Estrostep (ethinyl estradiol–norethindrone), and Yaz or Beyaz (ethinyl estradiol–drospirenone) [74].

Isotretinoin

Isotretinoin is a systemic retinoid indicated to treat scarring disease, severe nodulocystic acne, and cases in which oral antibiotics or hormonal therapies resulted in less than 50% improvement after four months.

It is considered to be highly effective because it targets all the factors that cause acne. It reduces sebum production by approximately 70%, normalizes follicular keratinization, reduces colonization by *C. acnes*, and exhibits anti-inflammatory properties [81].

As an agent with significant efficacy, the use of isotretinoin requires regular monitoring and adherence to the recommended guidelines for a safe and successful treatment [81].

### 3.3. Cannabinoids as Therapeutic Agents for Acne Treatment—A New Alternative

Although topical and systemic therapies are effective, as previously referred, they have some significant adverse effects that contribute to low patient compliance and adherence [88]. These side effects can result in patients discontinuing their treatment prematurely. For example, topical retinoids can cause skin irritation, resulting in dryness, burning, peeling, itching, erythema, and redness [77]. Topical retinoids have also been linked to a higher risk of photosensitivity, which can further elevate the risk of sunburn. Therefore, it is important to monitor the frequency of application and consider reducing it to mitigate these effects. While higher concentrations of retinoids may offer greater efficacy when used alone, they also have more pronounced side effects [77].

Regarding topical and oral antibiotics, their prolonged use can lead to bacterial and antimicrobial antibiotic resistance. It is also recommended to limit the use of systemic antibiotics according to the documented connections with inflammatory bowel disease, pharyngitis, and the development of Candida vulvovaginitis [77].

The common side effects of isotretinoin have also been extensively documented and reviewed. These side effects, which frequently resemble hypervitaminosis A symptoms, primarily affect the musculoskeletal, mucocutaneous, and ophthalmic systems. Additionally, there are concerns about potential adverse effects such as depression, anxiety and mood changes, cardiovascular risk factors, inflammatory bowel disease, bone mineralization, scarring, and *Staphylococcus aureus* (*S. aureus*) colonization [77]. Another set of side effects

may include dry eyes, lips, and nose, dermatitis, and the expression of embryotoxic and teratogenic properties [71,89–91].

The potential side effects associated with conventional acne treatments can discourage patients from initiating or continuing their therapy, which reinforces the need for new and safer therapies [71]. Therefore, in the last years, there has been a growing interest in exploring natural and plant-derived ingredients, leading to the discovery and development of new products that offer efficiency while causing less irritation, resulting in improved outcomes and adherence [88].

In recent research, particular attention has been given to the role of cannabinoids in the treatment of various skin disorders, considering the significant function of the ECS in skin health. In this regard, CBD has emerged as a promising acne treatment alternative, showing potential advantages in managing this skin condition [88].

CBD is a naturally occurring, non-psychoactive compound derived from the *Cannabis sativa* plant. It belongs to a class of over 100 phytocannabinoids present in the plant [71]. As the second major pharmacologically active component in hemp, CBD has garnered considerable interest for its potential therapeutic applications [1]. CBD has anti-inflammatory and immunomodulatory properties [71]. Thus, CBD has emerged as a potentially and promising therapeutic alternative for acne treatment due to its multifaceted actions.

### 3.3.1. Preclinical Findings on the Effect of Cannabinoids in Acne

Within the domain of acne research, preclinical investigations have revealed intriguing findings regarding the impact of cannabinoids. These studies delve into the potential therapeutic effects of cannabinoids on acne, exploring their influence on critical factors such as lipogenesis, inflammation, and microbial triggers.

#### Lipostatic Effects

Several preclinical findings regarding the effect of cannabinoids on acne have been reported. In a study conducted by Oláh et al. (2014) [92], CBD performed a unique "trinity of cellular anti-acne actions", emphasizing its potential in this regard. In this study, the authors used human sebocytes and human skin organ cultures (hSOC) to replicate the function of sebaceous glands in vivo and evaluate the effects of CBD. The study revealed that CBD had the ability to normalize the lipogenesis of sebocyte cells through a lipostatic effect without compromising the viability of the cells. Lipogenesis refers to the metabolic pathway through which these cells produce and accumulate lipids in order to produce sebum [92].

Additionally, this study demonstrated that the lipostatic effects of CBD were mediated through the activation of the TRPV4 ion channels, which resulted in an increase in $Ca^{2+}$ concentration [92]. This "negative regulation" of lipogenesis, achieved by the elevation of $Ca^{2+}$ concentration, was consistent with previous findings in sebocytes [93] and adipocytes [94,95].

Moreover, CBD has demonstrated the ability to effectively suppress the lipogenic effects of AEA [92]. This inhibition of lipogenesis is achieved by blocking the pro-lipogenic ERK1/2 MAPK pathway that is typically induced by AEA. The mechanism underlying CBD's inhibitory action involves the activation of the TRPV4 ion channel [92]. Notably, gene expression studies have shown that this activation leads to the downregulation of specific genes associated with lipid synthesis, with a particular focus on Nuclear Receptor Interacting Protein-1 (NRIP1), which is known to play a significant role in influencing glucose and lipid metabolism. This inhibition of sebocyte lipogenesis is a key factor contributing to the overall lipid regulatory effects of CBD [92].

In 2016, the same research group investigated the effects of other phytocannabinoids, specifically, cannabichromene (CBC), cannabidivarin (CBDV), cannabigerol (CBG), cannabigerovarin (CBGV), and tetrahydrocannabivarin (THCV), on human sebocyte functions [96]. Regarding basal lipogenesis (lipid synthesis), the study revealed that CBC and THCV inhibited this process, while CBG and CBGV increased it, making them potentially

pro-acne. Additionally, CBC, CBDV, and THCV demonstrated a reduction in AA-induced 'acne-like' lipogenesis [96].

According to the results of the study, CBC and CBDV displayed significant lipostatic activity, but THCV emerged as the most promising anti-acne compound [96]. Similar to CBD, THCV demonstrated universal lipostatic actions, completely counteracting 'acne-mimicking' activity induced by substances acting through independent pro-lipogenic signaling pathways. These included the effects of PKCd-activating AA, as well as AEA and 2-AG, primarily targeting the CB2R [96].

The results of the study [96] revealed a functional heterogeneity among the phytocannabinoids (pCBs) tested. While CBC, CBDV, and THCV exhibited a consistent pattern, CBG and CBGV behaved in an 'endocannabinoid-like' manner, enhancing sebaceous lipid synthesis in sebocytes. This suggests the potential utility of CBG and CBGV in managing conditions such as dry-skin syndrome, xerosis, and even skin aging [96].

Furthermore, additional preclinical investigations have demonstrated the potential anti-acne effects linked to the hexane extracts derived from hemp seeds (*Cannabis sativa* L.) [97]. Jin and Lee et al. (2018) [97] conducted an in vitro study using human HaCaT keratinocytes to assess, among other effects, the antilipogenic properties of the hemp seed hexane extracts [97]. This study specifically examined their effects on inflammation and lipogenesis induced by Propionibacterium acnes. Notably, these extracts exhibited antilipogenic properties when tested in insulin-like growth factor-1 (IGF-1)-stimulated lipogenesis, indicating their potential to mitigate excessive lipid production, a contributing factor in acne development [97].

In the investigation of antilipogenic effects, cultured human sebocytes were treated with IGF-1 to stimulate lipid production [97]. The results revealed that the hemp seed hexane extracts (HSHEs) suppressed intracellular lipid synthesis by enhancing the expression of phosphorylated AMP-activated protein kinase alpha (p-AMPKα) while reducing the levels of phosphorylated mammalian target of rapamycin (p-mTOR), PPARγ, sterol regulatory element-binding protein 1 (SREBP1), and fatty acid synthase (FAS) [97].

Moreover, HSHE influenced intracellular lipogenesis through the AKT/FoxO1 signaling pathway. In IGF-1-treated sebocytes, HSHE inhibited the phosphorylation of protein kinase B (AKT) and fork head box protein O1 (FoxO1), leading to a down-regulation of intracellular lipid production. Additionally, HSHE demonstrated a significant inhibitory effect on 5-lipoxygenase, a catalyst for leukotriene-B4 production, associated with inflammation and lipid synthesis in sebaceous glands. Treatment with 0.3% HSHE resulted in approximately 73% inhibition of 5-lipoxygenase compared to the group treated with IGF-1 alone. Therefore, HSHE demonstrated potential to regulate lipid synthesis in sebaceous glands by reducing the 5-lipoxygenase levels [97].

Antiproliferative Effects

Antiproliferative effects are also crucial in the treatment of acne vulgaris since they target the excessive growth and multiplication of certain cells, specifically sebocytes. These effects focus on the rapid division and multiplication of sebocytes, thereby reducing the overproduction of sebum. Compounds with antiproliferative properties, including cannabinoids like CBD, can modulate sebocyte activity and inhibit their excessive growth. This modulation of sebocyte proliferation aids in restoring a balance in sebum production, a key factor in acne management.

In a study conducted by Oláh et al. (2014) [92], CBD exhibited an antiproliferative effect on sebocyte cells, reducing their excessive cell growth without inducing apoptosis. The antiproliferative effects of CBD were proved to be mediated through the activation of the TRPV4 ion channels. This study demonstrated that CBD had a significant antiproliferative effect on human sebocytes, both in vitro and ex vivo. Therefore, it is expected that the observed effects of CBD will effectively lead to a reduction in sebum production in vivo, since sebum production is controlled by a process called holocrine secretion. This process is intricately tied to the vital roles played by the proliferative activity of basal layer sebocytes

located in the sebaceous gland, along with their integral participation in the synthesis of lipids, both of which bear significant importance [92].

In a study conducted by the same research group, in 2016, the authors found that not only CBD but also CBC, CBDV, CBG, CBGV, and THCV could induce cell death in sebocytes when tested in vitro at high concentrations ($\geq$50 μM) [96]. Furthermore, the same authors discovered that THCV inhibits the proliferation of sebocytes. They discovered that THCV, at concentrations equal to or less than 10 μM, demonstrated a dose-dependent reduction in cell proliferation over a 72 h treatment period without causing cytotoxic effects. Moreover, at its highest concentration tested, THCV completely stopped proliferation, showcasing significant sebostatic (lipostatic and antiproliferative) activity in vitro [96].

Anti-Inflammatory Effects

In a study conducted by Oláh et al. (2014) [92], CBD exhibited anti-inflammatory properties by decreasing the levels of pro-inflammatory cytokines [92]. CBD's anti-inflammatory activity, as demonstrated in the abovementioned study, was proven to be a TRPV4-independent process.

CBD effectively prevented an elevation in proinflammatory cytokine levels that are induced by toll-like receptor (TLR) activation or the action of other "pro-acne" agents [92]. This phytocannabinoid exerted complex anti-inflammatory effects via A2a adenosine-receptor-dependent up-regulation of tribbles homolog 3 (TRIB3) and inhibition of NF-kB signaling [92]. CBD can modulate adenosine signaling by inhibiting the rapid cellular uptake of adenosine via nucleoside transporter. Adenosine is a purine nucleoside that demonstrates protective attributes and attenuates inflammation through its activation of the A2a G-protein-coupled adenosine receptor. CBD promotes endogenous adenosine signaling as a protective mechanism during inflammatory and immune responses by inhibiting intracellular adenosine uptake [71].

In a study conducted by Oláh et al. (2016) [96], all the phytocannabinoids tested, namely, cannabichromene, cannabidivarin, cannabigerol, cannabigerovarin, and tetrahydrocannabivarin, exhibited anti-inflammatory properties, suggesting their potential for managing inflammatory skin diseases [96].

The effectiveness of the phytocannabinoids, including CBD, CBC, CBCV, CBDV, Δ8-THCV, Δ8-THC, and THC, in reducing inflammation was also demonstrated in a study conducted by Tubaro et al. (2010) [98] through topical application. The inhibition of the edematous (swelling) response served as a measure in the Croton oil mouse ear dermatitis test. The Croton oil mice ear assay is an in vivo test used to evaluate the anti-inflammatory properties of different substances, enabling the distinction between their specific profiles [98].

In a study conducted by Jin and Lee et al. (2018) [97], they also evaluated the anti-inflammatory properties linked to hemp seed hexane extracts [97]. This study specifically examined their effects on inflammation and lipogenesis triggered by Propionibacterium acnes. The extracts achieved anti-inflammatory effects by reducing the expression of genes encoding inflammatory enzymes, such as inducible nitric oxide synthase (iNOS) and cyclooxygenase-2 (COX-2), and inflammatory cytokines, like interleukins IL-1β and IL-8, and additionally through regulating the nuclear factor kappa B (NF-KB) and mitogen-activated protein kinase signal pathways [97].

In alignment with these previous findings, Jiang et al. (2022) discovered that CBD effectively reduces inflammation caused by extracellular vesicles of *C. acnes* [99], and, consistent with this finding, another study conducted by Perez et al. (2022) showed that both CBD and CBG inhibited the release of IL-8 and IL-1 from normal human epidermal keratinocytes when they were exposed to *C. acnes* [100].

Antimicrobial Effects

Antimicrobial effects are also vital in acne vulgaris treatment because they directly address the bacterial component, particularly *P. acnes*, responsible for the inflammatory cascade and exacerbation of acne lesions. By controlling microbial overgrowth, acne

treatments can effectively contribute to the overall management and improvement of this dermatosis.

In a study conducted by Jin and Lee et al. (2018) [97], the authors demonstrated that hemp seed hexane extracts have antimicrobial activity effects against *P. acnes*, i.e., the bacteria associated with acne. In this study, different concentrations of HSHE (0%, 15%, 20%, and 25%) were tested against *P. acnes*, and the number of colonies was counted to assess the antimicrobial activity. Results revealed that 15% and 20% HSHE exhibited approximately 59% and 99% antimicrobial activity, respectively, compared to the control. Notably, complete suppression of *P. acnes* growth was achieved at the 20% HSHE concentration. In contrast, erythromycin (3 ppm), a conventional antimicrobial agent for acne, demonstrated roughly 67% antimicrobial activity. These results displayed HSHE's potential to effectively inhibit *P. acnes* growth [97].

In addition, Blaskovich et al. (2021) demonstrated the selective activity of CBD against a specific group of Gram-negative bacteria while exhibiting a broader spectrum of activity against Gram-positive bacteria, including *Cutibacterium acnes* [101]. CBD exhibited a consistent minimum inhibitory concentration (MIC) ranging from 1 to 4 $\mu$g mL$^{-1}$ against a broad spectrum of over 20 Gram-positive bacteria types in the study. This included various strains of crucial pathogens, such as methicillin-resistant *Staphylococcus aureus* (MRSA), multidrug-resistant (MDR) *Streptococcus pneumoniae*, *Enterococcus faecalis*, as well as anaerobic bacteria, like *Clostridioides difficile* and *Cutibacterium acnes*. Furthermore, the researchers demonstrated that CBD does not induce resistance upon repeated exposure. This combination of properties presents a compelling case for further investigations into this underexplored class of compounds [101].

Overall, these pre-clinical findings highlight the specific properties of cannabinoids that are worth exploring further for the treatment of acne vulgaris. CBD, particularly, stands out due to its diverse array of significant properties, including its lipostatic, antiproliferative, antimicrobial, and anti-inflammatory effects.

### 3.3.2. Clinical Trials with Cannabinoids

The preclinical data mentioned above has prompted a multitude of ongoing clinical studies investigating the potential impact of cannabinoids on acne [52]. Cannabinoids appear to have the advantage of being relatively safe. Studies have reported and confirmed the transdermal penetration of phytocannabinoids, suggesting that these compounds can be effectively applied to the skin via topical pharmaceutical preparations such as creams [58]. This indicates that phytocannabinoids have the potential to be effectively utilized for treating acne and other dermatological conditions.

For example, Ali et al. (2015) [102] conducted a single-blind, comparative study involving eleven participants over a period of 12 weeks. In this study, a 3% cannabis seed extract cream was applied twice daily to the right cheek, while a control cream was applied to the left cheek. The results indicated a significant decrease in sebum levels and erythema on the right cheek where the cannabis seed extract cream was applied compared to the left cheek. No irritant or allergic reactions were observed, confirming the safety of the cream and suggesting the potential utility of this formulation in treating conditions like acne and seborrhea [102].

A phase 2 clinical trial reported on the ClinicalTrials.gov, involving 368 patients, evaluated the potential of CBD in a topical solution called BTX 1503, which contained synthetic CBD [103]. The aim of this research was to evaluate the safety and effectiveness of different doses of BTX 1503 liquid formulation in individuals experiencing moderate to severe acne vulgaris on their facial skin. The formulations tested were: BTX 1503 5% CBD (*w/w*) solution which was applied twice daily; BTX 1503 5% CBD (*w/w*) solution which was applied once daily; and BTX 1503 2.5% CBD (*w/w*) solution which was applied once daily. The results of this clinical trial were promising, demonstrating a significant reduction of 40% in acne lesions after 12 weeks of treatment. The administration of various doses of CBD in the study revealed that all tested doses were found to be safe, with no observed

adverse effects. The BTX 1503 5% CBD ($w/w$) solution applied once daily showed the best performance [103].

Moreover, Cohen et al. recently conducted a study to investigate the synergistic effects of natural plant extracts in combination with CBD, with the goal of treating acne by targeting diverse pathogenic factors while mitigating potential side effects [88]. The extracts tested were Centella asiatica triterpene (CAT) extract and Silybum marianum fruit extract. The outcomes revealed that the CAT extract, alongside silymarin extracted from Silybum marianum fruit, exhibited significantly improved anti-inflammatory efficacy when synergistically combined with CBD, surpassing the individual effects of each component. Furthermore, the CAT extract was found to enhance the inhibition of *C. acnes* growth induced by CBD. These three components were integrated into a topical formulation and evaluated using ex vivo human skin organ cultures. The formulation demonstrated both safety and effectiveness, leading to a reduction in the overproduction of IL-6 and IL-8 without compromising the viability of the epidermis [88]. Subsequently, a preliminary clinical trial involving 30 human subjects was conducted using this formulation. The results exhibited a statistically significant decrease in acne lesions, particularly those of an inflammatory nature, as well as lower levels of porphyrin. This outcome underscores a robust correlation between the findings obtained from in vitro, ex vivo, and clinical assessments [88].

Overall, cannabinoids have demonstrated a relatively safe profile in the studies conducted so far. Nevertheless, it is essential to conduct further research to determine the long-term safety and potential side effects of cannabinoid-based acne treatments. To establish the optimal formulations, dosages, and treatment regimens for cannabinoids in acne management, more extensive clinical trials are necessary. Moreover, exploring the potential benefits of other cannabinoids beyond CBD and their use in combination with current treatments (i.e., in combination therapy) to enhance efficacy and reduce side effects would be valuable.

The growing availability of cannabis-based products and their applications presents new opportunities for advanced routes of administration and delivery systems. Nanotechnology, for example, offers a promising avenue to enhance both the solubility and physicochemical stability of cannabinoids. The potential benefits of nanosystems, such as liposomes, nanoparticles, and micelles, to enhance CNB topical administration include increased efficacy and bioavailability, reduced toxicity, and controlled delivery. Although topically applying cannabinoids avoids first-pass metabolism [104], challenges such as the compounds' high lipophilicity, low water solubility [105], limited skin diffusion [106], and susceptibility to degradation from temperature, light, and auto-oxidation [107] need to be addressed. Further research into these delivery systems shows that they have the potential to significantly improve the efficacy of topical CNB [1,108].

CBD-based oils and creams applied topically typically exhibit a relatively low skin permeation/penetration rate, prompting the development of nanoemulsions (NEs) [109]. For instance, a study conducted by Lewinska et al. (2021) [110] demonstrated that NEs obtained through high-pressure homogenization maintained a stable particle size close to 200 nm for 30 days at 25 °C. These NEs showed no cytotoxicity to human skin cell lines, HaCaT keratinocytes, and normal human dermal fibroblasts, with cell viability values exceeding 80%. Moreover, the formulations exhibited a positive effect on skin hydration in human assays. Although cutaneous permeation studies were not conducted, the observed increase in skin hydration could potentially facilitate the passive transport of cannabidiol to deeper skin layers [110]. Therefore, NEs represent an alternative that could enhance both the topical and oral bioavailability of cannabinoids in different treatments [109]. Another alternative is polymer-based colloidal systems, such as polymer nanoparticles, which offer the advantage of providing sustained drug release and greater stability for the encapsulated material. The release rate of cannabinoids can be adjusted based on the type of polymer used, as different polymers present varying degradation mechanisms [109]. Investigating additional pathways to enhance cannabinoid activity is valuable, given the

diverse strategies available for improving their effectiveness and delivery. Nonetheless, it is essential to develop more translational studies in order to validate all the benefits observed in the in vitro assays [109].

In conclusion, it is essential to wait for further research and clinical validation before making definitive conclusions about the efficacy and widespread use of cannabinoids in acne treatment. Nonetheless, the emerging scientific evidence suggests that cannabinoids hold significant potential as a novel therapeutic option for acne and merit further investigation.

### 3.4. Legislation on Cannabis-Derived Ingredient Use

Despite the scientific breakthroughs, the landscape of cannabinoid use is significantly constrained by existing legislation, which is currently under development. The legislative landscape surrounding cannabis is constantly evolving on a global scale. Cannabis plants and their derivatives are subject to mandatory regulation under international laws, with specific provisions made for medical, cosmetic, and industrial applications [111].

The control of cannabis for medicinal purposes is governed by three key United Nations (UN) conventions, namely, the 1961 Single Convention on Narcotic Drugs (amended by the 1972 Protocol), the 1971 Convention on Psychotropic Substances, and the 1988 Convention Against Illicit Traffic in Narcotic Drugs and Psychotropic Substances. These conventions establish a comprehensive framework for the control and regulation of cannabis use [112].

In the European regulatory environment, the regulation of CBD is characterized by ambiguity and heterogeneity. There is no standardized legislation concerning the use of CBD within the European Union (EU). This can be attributed to various factors, including the reduced risk of CBD dependence and abuse, as well as its potential benefits for specific pathologies. Moreover, the absence of an explicit listing of CBD in the United Nations convention tables further contributes to the regulatory ambiguity in Europe [1,113].

Consequently, numerous CBD products claiming medicinal properties, including capsule supplements for various ailments and cosmetics like hemp oils, are being produced and distributed without proper regulatory supervision. These products often make health benefit claims without sufficient evidence and may contain unverified ingredients or components [114,115]. Therefore, it is essential for manufacturers, regulators, and consumers to remain vigilant and ensure compliance with evolving regulations to ensure the safety and efficacy of cannabinoid-based cosmetic and medicinal products.

For any medicinal product containing CBD to be sold or distributed in an EU Member State, it must obtain a community marketing authorization from the European Agency for the Evaluation of Medicinal Products (EMA) in accordance with Article 3 of Regulation (EEC) No. 2309/93. This marketing authorization will be applicable across the entire community [116].

Conversely, if a community marketing authorization is denied, it will result in a prohibition of the medicinal product's placement on the market throughout the community, as stipulated in Article 12 of Regulation (EEC) No. 2309/93 [116].

In recent years, there has been a notable increase in the accessibility of cannabis-based products, such as herbs, hemp, and oils containing CBD, commonly referred to as "light cannabis". All extracts of *Cannabis sativa* L. and any product that includes CBD, whether obtained synthetically or naturally, are categorized as novel foods, and must be subjected to strict control measures under the novel food regulation ((EU) 2015/2283). As in the case of medicinal products, novel foods require community authorization for supply in EU Member State markets, which is released by the European Food Safety Authority (EFSA) [116].

Regarding cosmetic products containing CBD, they can be used and placed on the EU market if they are derived specifically from the seeds and leaves of the cannabis plant, excluding the fruiting tops. To be considered permissible, however, these products must meet the criteria of being "safe for human health when used under normal or reasonably

foreseeable conditions of use", as outlined in Articles 3 and 4 of Chapter II of the Cosmetic Regulation (Regulation (EU) No 1223/2009). Additionally, these cosmetic products must not contain any substances listed in Annex II of this regulation [116].

Overall, the regulatory framework surrounding the utilization of products derived from *Cannabis sativa*, including topical formulations for both cosmetic and medical purposes, is currently undergoing development and lacks definitive establishment.

### 3.5. Available Therapeutics

Presently, the market offers a restricted range of cannabis-derived medications (as detailed in Table 3), with none of these specifically approved to treat dermatological conditions [1]. These formulations have received FDA approval and are only available with a prescription from a licensed healthcare provider [117]. They include Epidyolex®, Sativex®, Marinol®, Syndros®, Cesamet®, and Canemes®.

**Table 3.** Summary of authorized cannabis-based medicines, adapted from Martins et al. [1].

| Brand Name | Active Ingredient | Description | Indications | Dosage Forms | Countries Approved |
|---|---|---|---|---|---|
| Sativex® | Nabiximols | Plant based: THC/CBD (~1:1) | Spasticity and symptomatic relief of neuropathic pain in multiple sclerosis | Oro-mucosal spray | UK, Norway, other EU countries, Canada |
| Marinol® Syndros® | Dronabinol | Synthetic THC | Treatment of nausea and vomiting due to chemotherapy, anorexia due to AIDS | Gelatin capsules (Marinol®), oral solution (Syndros®) | USA, EU countries, Canada, others |
| Cesamet® Canemes® | Nabilone | Synthetic cannabinoid similar to THC | Treatment of nausea and vomiting due to chemotherapy in cancer patients; chronic pain management | Capsules | USA, Canada, some EU countries |
| Epidyolex® (EU) Epidiolex® (USA) | CBD | Purified CBD | Seizures associated with Lennox–Gastaut syndrome and Dravet syndrome | Oral solution | EU, USA |

Epidyolex® is a cannabis-based medicine containing a purified CBD formulation. It is primarily prescribed for managing seizures associated with Lennox–Gastaut syndrome and Dravet syndrome in patients aged 1 year and older (1). This oral medication contains 100 mg of CBD, which was meticulously extracted from the leaves and flowers of *Cannabis sativa* L. Complementary to its core formulation are excipients such as ethanol, sesame oil, and benzyl alcohol [117].

Sativex® represents another oral medicine. It contains 2.7 mg of Δ9-THC and 2.5 mg of CBD from *Cannabis sativa* L., and its excipients are ethanol and propylene glycol. The phytocannabinoids previously mentioned are extracted by liquid carbon dioxide from cannabis leaves and flowers. Sativex® is indicated for treating spasticity and providing symptomatic relief of neuropathic pain in multiple sclerosis [117].

Marinol® and Syndros®, both oral medications, are constituted by a synthetic cannabinoid called dronabinol that is similar to Δ9-THC. Marinol® contains dronabinol in doses of 2.5, 5, and 10 mg of active substance. Syndros® contains dronabinol in doses of 5 mg at a maximum. The first one has gelatin, glycerin, sesame oil, and titanium dioxide as excipients. The second one has dehydrated alcohol and propylene glycol as excipients. Both are approved for patients with acquired immunodeficiency syndrome (AIDS) and patients suffering anorexia related to weight loss, nausea, and vomiting induced by chemotherapy [117].

Cesamet® and Canemes® are composed of a synthetic cannabinoid, nabilone, in a dose of 1 mg, which is an analogue of Δ9-THC for oral administration. Their excipients are povidone and corn starch. Cesamet® and Canemes® are indicated for treating side effects of chemotherapy, such as nausea and vomiting [117].

In addition to these approved medications, there are numerous cannabis-derived and/or CBD products on the market; however, their effectiveness and safety have either not been accessed by regulatory agencies or their assessment remains undisclosed to the public [117].

Regarding cosmetic products, there are different CBD-based cosmetic products available on the market, such as creams, serums, cleansers, masks, and others, specifically intended for acne-prone skin and other related skin concerns. These products often claim to reduce inflammation, soothe irritated skin, and help balance oil production, which are beneficial for individuals with acne-prone skin.

Nordic Oil, for example, is a CBD brand that was founded in Scandinavia and offers a wide range of CBD products, including skincare items. Another notable brand is cbdMD, which includes a cream for acne that has been specially formulated to reduce blemishes and regulate the skin's moisture balance. In addition to these, other brands are exploring the potential of cannabinoids by integrating them into different products to leverage their beneficial properties. As time goes on, it is certain that there will be further progress in the development of products focused on addressing this skin condition.

## 4. Conclusions

The exploration of cannabinoids for acne treatment holds great potential, making it a valuable and interesting avenue for further research. Considering the high incidence of acne and the potential side effects associated with current treatments, cannabinoids appear to be a promising alternative. Moreover, the concept of sustainability is considered as these natural compounds offer eco-friendly options for skincare solutions.

The data presented in this review confirm that cannabinoids exhibit significant pharmacological properties for acne treatment. Literature reports have demonstrated the lipostatic, anti-inflammatory, and antiproliferative effects of cannabinoids. Among the cannabinoids studied, CBD stands out due to its broad anti-acne effects and its antipsychotic properties.

However, to fully understand the efficacy and optimal application of cannabinoids, including CBD, more extensive studies are necessary. The perceptible scarcity of data from clinical trials not only represents a limitation in current knowledge but also diminishes its practical utility for the reader. These studies should evaluate various concentrations, treatment durations, and formulations to determine the most effective and safe approach for utilizing cannabinoids in acne treatment. By continuing to investigate and refine this approach, we could potentially unlock a natural and sustainable solution for managing acne, benefiting both individuals and the environment.

**Funding:** This research received no external funding.

**Acknowledgments:** FCT—Fundação para a Ciência e a Tecnologia, I.P.—in the scope of the project UIDP/04378/2020 and UIDB/04378/2020 of the Research Unit on Applied Molecular Biosciences, UCIBIO, and the project LA/P/0140/2020 of the Associate Laboratory Institute for Health and Bioeconomy, i4HB.

**Conflicts of Interest:** The authors declare no conflict of interest.

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
