# Peer review of "Treatment Advances for Acne Vulgaris: The Scientific Role of Cannabinoids"

_cosmetics, doi:10.3390/cosmetics11010022_

Round 1

Reviewer 1 Report

Comments and Suggestions for Authors

The review on the potential therapy for acne through topical cannabinoids is certainly a valuable contribution to an interesting topic. However, the perceptible shortage of data from clinical trials not only represents a limitation tied to current knowledge but also diminishes its practical utility for the reader. Despite this challenge, the paper provides a comprehensive analysis of current research, presenting innovative perspectives. Furthermore, I would kindly like to point out that that there is an opportunity for optimization in terms of conciseness to improve reader accessibility.

Comments on the Quality of English Language

The English proficiency in the article is satisfactory, and the content is effectively communicated.

Author Response

Response: The reviewer’s positive feedback is greatly appreciated. We took suggestions as a valuable opportunity to improve the quality of our manuscript. Accordingly, a sentence based on the comments was added to the conclusions (lines 814-816).

Reviewer 2 Report

Comments and Suggestions for Authors

The manuscript has been well prepared, however there are some suggestion for improvement:

1. Lack of information on potential of Cannabis, eg. which countries are among the most producers? what number is the production?

2. In regards to the phytocannabinoids, are they originally only from Cannabis? please add about this

3. For the treatment using cannabinoids, in which form of the cannabinoids? are there any modifications to enhance the activity?

4. Part 3.3.1. Preclinical findings --> It is better if classified according to the mechanism of action of the cannabinoid, eg. 3.3.1.1. Lipostatic, etc

Author Response

1.The Lack of information on potential of Cannabis, eg. which countries are among the most producers? what number is the production?

Response 1: We appreciate the time and effort in providing feedback on our manuscript. Accordingly, we have incorporated the information as suggested (lines 69-87).

2.In regards to the phytocannabinoids, are they originally only from Cannabis? please add about this

Response 2: Thanking you for your comment. The suggested information was added (lines 111-117).

3.For the treatment using cannabinoids, in which form of the cannabinoids? are there any modifications to enhance the activity?

Response 3: Thank you for your comment. Accordingly, we added the suggested information (lines 666-671 and 677-695).

4.Part 3.3.1. Preclinical findings --> It is better if classified according to the mechanism of action of the cannabinoid, eg. 3.3.1.1. Lipostatic, etc

Response 4: Thank you for your suggestion. The section 3.3.1., related to preclinical findings, has been rewritten and subdivided into sections according to the mechanism of action of the cannabinoid.
